Massively parallel read mapping on GPUs with the q-group index and PEANUT

Köster Johannes johannes.koester@uni-due.de
Rahmann Sven
Genome Informatics, Institute of Human Genetics, University Hospital Essen, University of Duisburg–Essen , Germany
Kumar Abhishek
Electronic publication date: 2014 Sep 30
Publication date: 2014
Volume: 2
Electronic Location ID: e606
Received 2014 Jul 23; Accepted 2014 Sep 11
Copyright: © 2014 Köster and Rahmann
Copyright year: 2014
Copyright holder: Köster and Rahmann
License: This is an open access article distributed under the terms of the Creative Commons Attribution License, which permits unrestricted use, distribution, reproduction and adaptation in any medium and for any purpose provided that it is properly attributed. For attribution, the original author(s), title, publication source (PeerJ) and either DOI or URL of the article must be cited.
License URL: https://creativecommons.org/licenses/by/4.0/

Keywords: Read mapping, Datastructure, Parallel algorithms, GPU programming

Funding: German Research Foundation DFG SFB 876/C1 Collaborative Research Center (Sonderforschungsbereich, SFB) Mercator Foundation MERCUR Pe-2013-0012 Part of this work was funded by the German Research Foundation (DFG), Collaborative Research Center (Sonderforschungsbereich, SFB) 876 “Providing Information by Resource-Constrained Data Analysis” within subproject C1, see http://sfb876.tu-dortmund.de. Sven Rahmann acknowledges support from the Mercator Foundation for project MERCUR Pe-2013-0012 (UA Ruhr Professorship “Computational Biology”). The funders had no role in study design, data collection and analysis, decision to publish, or preparation of the manuscript.

==============================
We present the q-group index, a novel data structure for read mapping tailored towards graphics processing units (GPUs) with a small memory footprint and efficient parallel algorithms for querying and building. On top of the q-group index we introduce PEANUT, a highly parallel GPU-based read mapper. PEANUT provides the possibility to output both the best hits or all hits of a read. Our benchmarks show that PEANUT outperforms other state-of-the-art read mappers in terms of speed while maintaining or slightly increasing precision, recall and sensitivity.

Introduction

A key step in many next generation sequencing (NGS) data analysis projects, e.g., for variant calling or gene expression analysis, is mapping the obtained DNA reads to a known reference (e.g., the human genome). The read mapping problem is to find the possible origins of each read within the reference and to optionally provide basepair-level alignments between each read and each originating region. A read mapping algorithm has to be error tolerant because of mutations and sequencing errors. The optimal solution to this problem is to calculate alignments of each read against the reference. Since this would incur an infeasible running time (proportional to the product of reference length and total read lengths, e.g., 6 Gbp⋅107⋅100 bp = 6⋅1018 bp2 for the human genome and its reverse complement against 10 million Illumina reads), various filtering methods and approximations have been developed. These can be roughly classified into methods based on backward search using the Burrows–Wheeler transform (BWT; Burrows & Wheeler (1994)), e.g., BWA (Li & Durbin, 2009) or Bowtie 2 (Langmead & Salzberg, 2012), and methods based on q-gram indexes, e.g., RazerS 3 (Weese, Holtgrewe & Reinert, 2012). Both techniques are used to quickly locate short exact matches as anchors for putative alignments. BWT-based techniques require less memory, especially together with compression techniques, but their usage requires more computation cycles. In contrast, q-gram index techniques use simple lookups and are easily implemented, but they have a larger memory footprint.

Among the best alignments of a read, it is sometimes not obvious which one represents the true origin of the read. In the following, the candidate origins of a read reported by a read mapper are called hits. Weese, Holtgrewe & Reinert (2012) categorize read mappers into best-mappers that try to find the (or any) best hit of a read (e.g., BWA-MEM; Li (2013)) and all-mappers that provide a comprehensive enumeration of all hits (e.g., RazerS 3 or the “all” mode of Bowtie 2) up to a given error threshold. While all-mappers can be much slower (depending on the number of hits), their strategy is advantageous when a confidence value is required that the reported origin is the true origin of the read. Further, all-mappers are useful when mapping to homologous sequences like alternative transcripts (Roberts & Pachter, 2013) or meta-genomes, where more than one hit is expected. An intermediate strategy is to report all hits of the best stratum, i.e., all hits with the same lowest error level (instead of only the first or a random such hit).

Recently, exploiting the parallelization capabilities of graphics processing units (GPUs) for read mapping has become popular and GPU-based BWT-read-mappers appeared, e.g., SOAP3 (Liu et al., 2012), SOAP3-dp (Luo et al., 2013) and CUSHAW2-GPU (Liu & Schmidt, 2014). Using a q-gram index on a GPU is not a common choice because of its large size. Therefore, to the best of our knowledge, q-gram index based mappers so far only use the GPU for calculating the alignments and keep the index on the CPU, e.g., Saruman (Blom et al., 2011) and NextGenMap (Sedlazeck, Rescheneder & von Haeseler, 2013).

Here, we present the q-group index, a novel data structure for read mapping which is a variant of the classical q-gram index with a particularly small memory footprint. The q-group index comes with efficient parallel algorithms for building and querying, targeted towards modern GPUs. To the best of our knowledge, the q-group index is the first feasible implementation of q-gram index functionality on the GPU. On top of the q-group index we present PEANUT (ParallEl AligNment UTility), a GPU-based massively parallel read mapper. PEANUT provides both an all-mapping and a best-mapping mode and is the first GPU-based all-mapper. With both a recent and a four years old NVIDIA™ Geforce GPU, we show that PEANUT outperforms other state of the art best-mappers and all-mappers. For all-mapping, PEANUT is 4–10 times faster. PEANUT shows a slightly higher precision and recall than other best-mappers and an improved sensitivity compared to other all-mappers at default parameters.

This article is structured as follows. We first discuss the GPU architecture and its implications for designing the q-group index to maximize parallel GPU usage (“Designing for efficient GPU usage”). Then, we describe the q-group index data structure (“The q-group index”) and present the PEANUT approach of read mapping with the q-group index (“The PEANUT algorithm”). The “Results” section shows benchmark results on speed, precision, recall and sensitivity of PEANUT. A brief discussion concludes the paper.

Methods

Designing for efficient GPU usage

We use the terminology of NVIDIA™, while the general concepts are also applicable to competitors like AMD™. A GPU is partitioned into streaming multiprocessors (SMs), each of which has its own on-chip memory, cache and processing cores. By adjusting the thread block size it can be controlled how threads are distributed among the SMs. One thread block is executed on one SM and continues resident until all threads in the block are completed. Once a thread block has finished, another will be scheduled to the SM if any blocks are left. An SM can execute 32 threads in parallel (restricting the thread block size to be a multiple of 32); such a group of threads is called a warp or wavefront. At any time, each of these threads has to execute the same instruction in the code, but may do so on different data, a concept which is called single instruction, multiple threads (SIMT). Hence, conditionals with diverging branches should be avoided, since threads taking an if-branch have to wait for threads taking the corresponding else-branch to finish and vice versa. All SMs may access a slow common global memory (about ≤3 GB on most of today’s GPUs) in addition to their fast on-chip cache and memory. While the size of the fast cache is extremely limited, accesses to the global memory are slow and should be minimized. However, the memory latency can be reduced by coalescing the access, i.e., letting threads in a warp access contiguous memory addresses, such that the same memory transaction can serve many threads. In addition, an SM can execute a different warp while waiting on a transaction to finish, thereby hiding the latency. For the latter, threads should minimize their register usage such that the number of warps that can reside on an SM is maximized. Finally, data transfers from the main system memory to the GPU global memory are comparatively slow. Hence it is advisable to minimize them as well.

A useful programming pattern that is used extensively in the algorithms presented in this work are parallel prefix scans, a special case of which is the computation of a cumulative sum, which at first appears to be a serial process, or filtering an index set. Parallel prefix scans are used to solve these problems in a data parallel way with a minimum amount of branching, thus nicely fitting the above considerations; Blelloch (1990) provides a comprehensive introduction. We use the PyOpenCL implementation of prefix scans (Klöckner et al., 2012).

The q-group index presented in this work is designed with above considerations in mind: The data structure supports simple access patterns that minimize the register usage. This allows to hide latency caused by not coalesced memory accesses. We evaluate this in “GPU resource usage”. The parallel algorithms presented in this work further minimize the data transfers necessary between their steps to constant amounts of data (e.g., single integers). Where appropriate, we discuss architecture specific considerations in the text.

The q-group index

A classical DNA q-gram index of a text T stores, for each string of length q over the alphabet, at which positions in T the q-gram occurs and allows retrieving these positions in constant time per position. It is commonly implemented via two arrays that we call the address table A and the position table P.

Q-grams are encoded as machine words of appropriate size with two subsequent bits encoding one genomic letter (i.e., A = 00, C = 01, G = 10, T = 11). Unknown nucleotides (usually encoded as N) are converted randomly to A, C, G or T, and larger subsequences of Ns are omitted from the index. Hence, a q-gram needs 2q bits in hardware and is represented (encoded) as a number g∈{0, …, 4q−1}. The address table provides for each (encoded) q-gram g a starting index A[g] that points into the position table such that P[A[g]], P[A[g] + 1], …, P[A[g + 1]−1] are the occurrence positions of g.

Deciding about the q-gram length q entails a tradeoff between specificity of the q-grams and the size of the data structure. Array A needs 4q integers and thus grows exponentially with q, while array P needs |T| integers, independently of q. Larger values of q lead to fewer hits per q-gram that need to be validated or rejected in later stages. Further, the choice of q determines the sensitivity or error-tolerance of the search via the pigeonhole principle (q-gram lemma): only if there are e < (n + 1)/q−1 errors, we can guarantee that a q-gram match exists.

Idea

The idea of the q-group index is to have the same functionality as the q-gram index (i.e., retrieve all positions where a given q-gram occurs in constant time per position), but with a smaller memory footprint for large q. This is achieved by introducing additional layers in the data structure. In the following, we always consider a q-gram as its numeric representation g∈{0, …, 4q−1}.

We divide all 4q q-grams into groups of size w, where w is the GPU word size (typically w = 32). The q-gram with the number g is assigned to group number ⌊g/w⌋. Thus the ith group is the set Gi = {g|⌊g/w⌋ = i} of w consecutive q-grams according to their numeric order. The set of all q-groups is Gq≔G0,G1,…,G⌈4qw⌉−1. We write gij for the jth q-gram in Gi.

For a given q and text T, the q-group index is a tuple of arrays IT,q≔I,S,S′,O.

Array I consists of |Gq| words with w bits each (overall 4q bits), and bit j of I[i] indicates whether gij occurs at all as a substring in the text, i.e., Iij=1if g is a substring of T,0otherwise.

The array O corresponds to the position table P of a regular q-gram index: it is the concatenation of all occurrence positions of each q-gram in sorted numeric q-gram order. To find where the positions of a particular q-gram g begin in O, we first determine the group index i and the q-gram number j within the group, such that g = gij. With the bit pattern of I[i], we determine whether qij occurs in T. If not, there is nothing else to do. If yes, i.e., I[i]j = 1, we determine the j′ such that bit j is the j′th one-bit in I[i].

The address array S contains, for each group i, an index into another address array S′, such that S′[S[i] + j′] is the starting index in O where the positions of gij can be found; see Fig. 1 for an illustration. All occurrence positions are now listed as OS′Si+j′,…,OS′Si+j′+1−1.

Figure 1 The q-group index.

The q-group index consists of four arrays I, S, S′, O. The purple arrows illustrate how the four layers of the index are traversed to reach the occurrences of the queried q-gram GAAA.

Similar to a plain q-gram index, access is in constant time per position: For a given q-gram g, to determine (i, j) such that g = gij, we simply compute i = ⌊g/w⌋ and j = g−wi = jmodw. To compute j′, i.e., find how many one-bits occur up to bit j in I[i], we use the population count instruction with a bit mask: j′=PopcountIi & 2j−1.

The population count Popcount(x) returns the number of 1-bits in x.

Construction

Algorithm 1 shows how the index is built. The outline of of the algorithm is as follows. First, I is created from the q-grams of the text (line 2). Then, S is calculated as the cumulative sum over the population counts of I (line 6). Next, the number of occurrences for each q-gram is calculated (line 10) and S′ is created as the cumulative sum over these counts (line 14). Finally, the q-gram positions are written into the appropriate intervals of O (line 16).

Each step is implemented on the GPU with parallel OpenCL1 kernels. The cumulative sums are implemented with parallelized prefix scan operations (see “Designing for efficient GPU usage”). Importantly, the algorithm needs hardly any branching (hence maximizing concurrency) and makes use of coalescence along the reads in order to minimize memory latency. All major data structures are kept in GPU memory.

Size

To determine the worst case size of the q-group index, we note that both I and S consist of ⌈4q/w⌉ words, S′ contains an index for each occurring q-gram and hence of min{4q, |T|} words in the worst case, and O is a permutation of text positions consisting of |T| words. Thus the q-group index needs up to 2/w⋅4q + min{4q, |T|} + |T| words, compared to a conventional q-gram index with 4q + |T| words. To evaluate how the worst case size of the q-group index behaves compared to the size of the q-gram index, we consider the ratio K between the possible q-grams and the text size, i.e., 4q = K|T|.

If 4q⪅|T|, the conventional q-gram index has a small advantage because each q-gram occurs (even multiple times). In fact, assuming w = 32, if K ≤ 1, the size ratio between q-group index and q-gram index is K/16+K+1|T|/K+1|T|=1+K161+K. For K = 1 (or |T| = 4q), this means a small size disadvantage of 3% for the q-group index.

If, on the other hand, q becomes larger for fixed text size (such that q-grams become sparse), the q-group index saves memory, up to a factor of 16: For K > 1 the size ratio is (K/16 + 1 + 1)|T|/((1 + K)|T|) = (2 + K/16)/(1 + K), which tends to 1/16 for large K. The break-even point is reached for K = 16/15.

In practice, we use q = 16 because a bit-encoded DNA 16-mer just fills a 32-bit word and q = 16 offers reasonable error tolerance and high specificity. In this regime, with current GPU memory size, we use |T| = 108 (processing 100 million nucleotides at a time), so the size ratio is K = 42.95, and the q-group index needs only 10% of the memory of the conventional q-gram index. Figure 2 shows the behaviour of the index size ratio depending on K.

Figure 2 Ratio between the worst case size of the q-group index and the size of the q-gram index.

Ratio between the worst case size of the q-group index and the size of the q-gram index for different K. The purple line marks the break-even point, i.e., the K beyond which the q-group index guarantees to be smaller than the classical q-gram index.

Without noticable increase in access time, we can reduce the memory usage of the q-group index further by sampling every second (say, even) position of S and adding another population count and addition instruction to the code for odd positions.

The PEANUT algorithm

PEANUT uses the filtration plus validation approach: It first quickly detects exact matches of a given length q between each read and each reference (so-called q-gram matches) and only computes alignment scores where such matches are found. In total, the PEANUT algorithm for read mapping consists of the three steps (1) filtration, (2) validation and (3) postprocessing. The first two steps, filtration and validation, are handled on the GPU, while postprocessing is computed on the CPU.

The steps are conducted on a stream of reads. Reads are collected until buffers of configurable size are saturated. Then, any computation is done in parallel for all buffered reads (see Fig. 3).

Figure 3 The PEANUT algorithm.

Read sequences are buffered and a q-group index is created from them on the fly. Filtration (detection of q-gram hits) and validation are performed on the GPU until all reference sequences are processed. The hits are postprocessed and streamed out in SAM format. All layers operate independently in parallel and communicate via queues. Blue layers are I/O bound, green is executed on the GPU, and purple is executed on the CPU. Arrows between the layers denote a data transfer via a queue.

In the filtration step, potential hits between the reference sequence and the reads are detected using the q-group index. Next, the potential hits are validated using a variant of Myers’ bit-parallel alignment algorithm (Myers, 1999; Hyyrö, 2003). The validated hits undergo a postprocessing that annotates them with a mapping quality and calculates the actual alignment. The postprocessed hits are streamed out in SAM format (Li et al., 2009). Because of memory constraints on the GPU, all steps are performed per chromosome instead of using the reference as a whole.

Filtration

The filtration step aims to yield a set of potential hits, e.g., associations between a reference position and a read. For this, we seek for matching q-grams (i.e., substrings of length q) between the reference and the reads.

First, a subset of reads is loaded into memory and its q-group index is created on the GPU on the fly (see Algorithm 1). If the index was built over the reference, either repetetive large data transfers or on-line rebuilding of the index would be necessary for each set of buffered reads and each chromosome. Hence—inspired by RazerS 3 (Weese, Holtgrewe & Reinert, 2012)—we build the q-group index over concatenated reads instead of the reference, so we only build the index once for each set of buffered reads.

We now explain how to use the q-group index to retrieve potential hits. As stated above, the filtration step will be executed for a buffered set of reads, separately on each reference chromosome. The reference positions of interest are streamed against the index, and hits are obtained by reference position.

We assume that there is a function Indexpair(g) that returns, for a q-gram g, an index pair (kstart, kend) such that the occurrence positions in the indexed text are all O[k] with kstart ≤ k < kend, where O is from the q-group index built with Algorithm 1. Given the q-group index (I, S, S′, O), the function Indexpair(g) is implemented as follows. Let (i, j)≔Group-And-Bit(g) and j′≔Grouprank(I, i, j). Then kstart = S′[S[i] + j′] and kend = S′[S[i] + j′ + 1].

Algorithm 2 shows how putative hits are generated by querying the q-group index of buffered reads for each q-gram of the reference. First, the number of hits per reference position are computed in parallel and stored in array C (loop in line 2). In the following, only positions with at least one hit are considered (line 5). The cumulative sum of the counts generates an interval for each position, that determines where its hits are stored in the output array of the algorithm (line 6). Finally the occurrences for each reference q-gram are translated into hits that are stored in the corresponding interval of the output array (loop in line 8). We translate the position inside the index of concatenated reads into a read number (line 12) and a “hit diagonal” that denotes the putative start of the read in the reference (line 13, see Rasmussen, Stoye & Myers (2006)).

Again, each step of Algorithm 2 is implemented on the GPU with parallel OpenCL kernels. The filtering of P (line 5) uses parallelized prefix scan operations (see “Designing for efficient GPU usage”). All data structures reside in GPU memory; between the steps, at most constant amounts of data have to be transferred between host and GPU (e.g., a single integer).

The set P of reference positions to investigate and the reference sequence are retrieved from a precomputed HDF5-based reference index.2 First, this speeds up access to the reference. Second, we omit exceedingly frequent q-grams from the set P, as these only yield uninformative hits when using q-gram index based algorithms. Finally, P is sorted in numerical order of the q-grams. This increases the memory coalescence when accessing the q-group index, since subsequent threads will have a higher probability to access the same region in the index and hence the same memory bank in the global GPU memory.

Validation

The validation step takes the potential hits of the filtration step and calculates the edit distance (with unit costs, i.e., the Levenshtein distance) between a read and the reference sequence at its putative mapping position. If the edit distance is small enough, the hit is considered to be correct and will be postprocessed in the next step.

The edit distance is calculated with Myers’ bit-parallel algorithm (Myers, 1999) that simulates the edit matrix E between reference sequence and read, which contains one column for each reference base and one row for each read base. The value Eij is the minimal edit distance between the read prefix of length i and any substring of the reference that ends at position j. Interpreted as a graph with a node for each matrix entry, a path in E from the top to the bottom row denotes a semi-global alignment between read and reference. Myers’ algorithm calculates the edit matrix column-wise. The state of the current column is stored in bit vectors. A transition from one column to the next is achieved via a constant amount of bit-parallel operations on the bit vectors. In iteration j, the minimal distance between the read and any substring of the reference that ends at position j can be retrieved. If the accepted error rate is limited, only a part of the edit matrix is needed to calculate the optimal edit distance. Our implementation of the algorithm follows a banded version that calculates only the relevant diagonal band of the edit matrix (Weese, Holtgrewe & Reinert, 2012). The implementation keeps the considered part of each column in a single machine word of size w (currently 32 bits). Thereby it provides a complexity of O|r| with |r| being the read length. While the reduction to the diagonal band restricts the maximum insertion or deletion size in a single alignment, mismatches are not affected. Hence, the procedure allows to discover partial matches of the read, as needed for split read mapping. Large indels can be rescued later when calculating the actual alignment if a sufficiently large portion of the read aligns in this step.

Similar to Weese, Holtgrewe & Reinert (2012) we use the algorithm to calculate the edit distance of a semi-global alignment in backward direction, thereby obtaining the best starting position of the alignment. For each hit, the fraction of matches or percent identity is obtained which we define here in compliance with RazerS 3 as (|r|−k)/|r|, where |r| is the read length and k is the edit distance. Hits with a percent identity less than a given threshold are discarded. The default for this threshold is 80 percent which provides a decent sensitivity in our benchmarks (see “Results”). Decreasing it has moderate impact on performance, since more hits have to be postprocessed and written to disk.

Postprocessing

The postprocessing of a read removes duplicate hits (generated by clusters of matching q-grams), sorts the remaining hits by percent identity, pairs mates in case of paired-end alignment, estimates a mapping quality and calculates the actual alignment of each hit. For the latter, either local or semi-global alignments can be chosen. In contrast to the previous steps, postprocessing is done on the CPU. This allows us to postprocess the hits in parallel to filtration and validation.

Intuitively, a particular hit is more likely to be the true origin of a read the fewer hits with the same or with a better score occur. During postprocessing, hits are sorted into strata of the same percent identity (Marco-Sola et al., 2012). In paired-end mode, the percent identities of properly paired hits are summed when determining strata. Upon invocation, PEANUT can be configured to discard hits based on their stratum, e.g., providing only the best stratum or all strata. In the following, we refer to these as best-stratum and all modes.

For the remaining strata, a confidence value for distinguishing true positives (i.e., hits referring to the true origin of a read) from false positives is needed. PEANUT reports alignments in the SAM (Li et al., 2009) format, which assesses this in terms of the mapping quality. For each hit, the mapping quality is expected to approximate the probability 1−Pr(p|r) that the hit position p is not the true origin of the read r in the reference. Li, Ruan & Durbin (2008) define Pr(p|r) in a Bayesian way as Prp|r=Prr|p∑p′∈PPrr|p

with P being the set of all reference positions and approximate the likelihood Pr(r|p) of a read r to be sampled from position p (in the following called the sampling likelihood) as product of the miscall probabilities of mismatching bases as obtained from the sequencer. In practice, Pr(p|r) is approximated roughly using the best and second best hit (Li, Ruan & Durbin, 2008; Li & Durbin, 2009; Li & Durbin, 2010; Liu & Schmidt, 2012). PEANUT however shall be able to provide mapping qualities for all hits in the extreme. In contrast to best-mappers, we have access to the percent identities of all hits down to a given threshold (see “Validation”). For any hit, we choose to approximate the sampling likelihood as follows. Each edit operation in the underlying (but unknown) alignment is either a substitution, insertion or deletion. If the alignment represents the true sampling position of the read, all three may occur either due to a genetic variation in the sequenced sample compared to the reference sequence, or due to a sequencing error. All cases are unlikely and dominated by the expected sequencing error rate of about 2%. Hence, the sampling likelihood decays exponentially in the number of edit operations in the alignment. Therefore, we approximate Prr|p≈Ce−λk

with k being the error rate of the alignment obtained as 100−s from the percent identity s∈[0, 100] of the hit. Per default, λ and C are set to 1. This is a rough but conservative and quite general approximation. Under the assumption that this estimate is almost 0 for hits discarded during validation (since they will have a small percent identity) we can approximate the posterior probability Pr(p|r) as Prp|r≈Prr|p∑p′∈P′Prr|p

with P′ being the validated hit positions of the read r. The PHRED-scaled mapping quality is then obtained as min{−10log10(1−Pr(p|r)), 60}. Per default, we cap the mapping quality at 60 and force it to 0 in case of ambiguous hits (i.e., two or more best hits with the same percent identity) in order to generate values comparable to other read mappers like BWA (Li & Durbin, 2009). This is useful to satisfy the expectations of downstream analysis steps (e.g., ambiguously mapping reads are often expected to have a mapping quality of 0). The section “Evaluation of mapping qualities” below evaluates the quality of the approximation.

Results

We evaluate PEANUT in terms of its efficiency of GPU resource usage, its accuracy and its speed. We also evaluate the ability of the mapping quality measure defined in “Postprocessing” to separate true hits from others.

GPU resource usage

In order to maximize utilization of the GPU hardware, idle cores have to be avoided. The two most important reasons for idle cores are branching and memory latency (see “Designing for efficient GPU usage”). The latter can be hidden if the SM is able to execute a different warp while waiting on the transaction. The capability to do so can be measured with the occupancy, that is the fraction of active warps among the maximum number of warps on an SM. The more active warps exist on an SM, the higher is the chance that latency can be hidden by executing another warp. Figure 4 shows the occupancy patterns of the implemented OpenCL kernels, as measured with the NVIDIA™ CUDA command line profiler depending on the used thread block size (see “Designing for efficient GPU usage”). The thread block size influences the occupancy by limiting the number of potentially active warps and determining the amount of used registers and shared memory on the SM. Since the latter are limited, a bigger thread block size does not necessarily lead to a higher occupancy. As can be seen, the occupancy for all steps is high. For building of the q-group index (create_queries_index) and the filtration step (filter_reference), it even reaches 1.0 which illustrates the benefit of the q-group index being tailored towards the GPU architecture.

Figure 4 Occupancy of GPU cores depending on the thread block size.

Shown are representative patterns for OpenCL kernels from the three main steps of the algorithm: building the index (create_queries_index), filtration (filter_reference) and validation (validate_hits).

Sensitivity

Here, we strive to evaluate the sensitivity of PEANUT in terms of its ability (and hence that of the q-group index) to detect all alignments up to a given error rate. For this, we use the Rabema (Holtgrewe et al., 2011) benchmark that allows to compare mapping results based on a formalized framework. With Rabema, the genomic origin of a read itself defines how tight a mapping has to be using equivalence classes of ambiguous alignments. Further, it supports fair benchmarking of both all-mapping and best-mapping. First, 10,000 Illumina3 reads of length 100 were simulated using the read simulator Mason (Holtgrewe, 2010), which is part of the Seqan project (Döring et al., 2008), with the Saccharomyces cerevisiae genome,4 default parameters and error rates. Second, the simulated reads were mapped to the genome using RazerS 3 (Weese, Holtgrewe & Reinert, 2012) with full sensitivity and different error tolerances. In this configuration, RazerS 3 guarantees to report all alignments of a read up to a given error rate. For above notion of sensitivity, it would not be sufficient to consider the true origins of a read as they are known from the simulation with Mason. The mapped reads were used to generate gold standards for Rabema to test against. In “Comparison with other read mappers”, we assess the accuracy of PEANUT in comparison to other mappers on larger datasets.

We analyze the sensitivity of our algorithm using q-grams of length 16, because it is computationally optimal on the current GPU hardware. We configure PEANUT to provide all semi-global alignments of a read and leave all other parameters at their default values. Sensitivity is assessed by the Rabema measure “Normalized found intervals” (Holtgrewe et al., 2011) and all alignments of a read are considered (all-mode). With a percent identity threshold of 60 (see “Validation”) our algorithm provides 100% sensitivity for error rates below 5%, 99.86% sensitivity for error rates up to 10% and still 98.86% sensitivity with an error rate up to an unrealistically high 20%. With a stricter threshold of 80, PEANUT still reaches 98.81% sensitivity for the latter.

In general, we advise to set the percent identity threshold to be slightly more permissive than the expected error rate. This is because the replacement of N-characters in the reads and the reference with random bases can introduce additional mismatches. Above rates are far better than the worst case sensitivity that can be expected by applying the pigeonhole principle (i.e., with reads of length 100 and q-grams of length 16, we can expect to find at least one perfectly matching q-gram for all alignments with at most 6 errors), such that using 16-grams appears to be a reasonable default choice in practice.

Comparison with other read mappers

We compare the PEANUT algorithm with other state of the art read mapping algorithms in terms of run time and accuracy. The evaluation is conducted on 4 datasets:

1. 5 million simulated Illumina HiSeq 2000 reads;

2. 5 million real Illumina HiSeq 2000 reads from the human exome;

3. 10 million real paired-end Illumina HiSeq 2000 reads from the human exome;

4. 50 million real paired-end Illumina HiSeq 2000 reads from the whole human genome.

The simulated reads (dataset 1) were created from the ENSEMBL human reference genome5 version 37 with Mason (see “Sensitivity”). The read length is set to 100 and all other parameters of Mason are left at their default values, such that reads with a typical error profile and mutation rate are generated. The second and third datasets are generated from real paired-end exome sequencing reads6 (Martin et al., 2013) of length 100 obtained from a patient suffering from uveal melanoma (a cancer of the eye) sequenced with an Illumina HiSeq 2000 sequencer. Dataset 2 consists of the first 5 million forward reads. Dataset 3 consists of both the first 5 million forward and backward reads, i.e., 10 million reads in total. Dataset 4 is generated from real paired-end whole genome reads of length 200 obtained from an african male.7 The reads are part of the Illumina Platinum Genomes.8 The first 25 million forward and backward reads were chosen, i.e., 50 million reads in total.

The benchmark was conducted on an Intel Core i7-3770™ system (4 cores with hyperthreading, 3.4 GHz, 16 GB RAM) with an NVIDIA Geforce™ 780 GPU (12 SMs, 3GB RAM) and a 7200 rpm hard disk. We evaluated two modes of PEANUT. First, PEANUT was configured to find the best stratum of semi-global alignments (best-stratum mode) for each read. Second, PEANUT was configured to find all semi-global alignments (all mode) for each read. For comparison, we benchmarked the newest generation of BWA (BWA-MEM, version 0.7.5; Li (2013)), Bowtie 2 (version 2.0.2; Langmead & Salzberg (2012)), CUSHAW 3 (version 3.0.3; Liu, Popp & Schmidt (2014)), CUSHAW2-GPU (version 2.1.8; Liu & Schmidt (2014)), NextGenMap (version 0.4.11; Sedlazeck, Rescheneder & von Haeseler (2013)), RazerS 3 (version 3.2; Weese, Holtgrewe & Reinert (2012)) and MrFast (version 2.6.0.1; Alkan et al. (2009)). All tools were configured to use 8 threads (the reasonable choice in case of 4 cores with hyperthreading). For MrFast, which does not support multithreading directly, this was achieved by partitioning the input files containing the reads into 100 equally sized chunks and running 8 parallel instances of MrFast with the Unix command parallel (the time for merging the resulting output was not included into the run time). NextGenMap was used in GPU mode, such that it makes maximum use of the available hardware. All read mappers were configured to output alignments in SAM format (Li et al., 2009) directly to the hard disk.

We outline the reasons for excluding several available read mappers from the benchmarks. At the time of writing (07/2014), no working installations of SOAP3 (Liu et al., 2012), SOAP3-dp (Luo et al., 2013) and BarraCUDA (Klus et al., 2012) could be obtained. A binary compiled against the NVIDIA CUDA setup of the test system (NVIDIA CUDA 6) was not available for SOAP3 and SOAP3-dp. The compilation of SOAP3-dp-r177 and SOAP3-r146 fails on our test system. The Barracuda read mapper compiles but refuses to run on CUDA 6. Finally, read mappers specialized on RNA-Seq (Dobin et al., 2013; Trapnell, Pachter & Salzberg, 2009) were excluded, as this exceeds the scope of PEANUT. In principle, STAR (Dobin et al., 2013) could be applicable to DNA reads, and the authors claim significant speedups compared at least to other RNA-Seq focused read mappers. However, this comes at the cost of extensive memory usage by an uncompressed suffix array, which exceeds the capacity of the used test system.

In the following, we distinguish between all-mappers and best-mappers (PEANUT occurs in both categories, using the all-mode and the best-stratum mode). Best-mappers only strive to find the single origin of a read on the reference sequence of a single organism. All-mappers provide all alignments of a read down to a given error rate. Hence, all-mapping is computationally more intensive.

Run time performance is measured three times as the overall wall clock time for processing a dataset on the test system. Table 1 shows the run times for PEANUT and its competitors on all datasets. First, PEANUT in best-stratum mode outperforms all best-mappers (including the other GPU based mappers NextGenMap and CUSHAW2-GPU) on all datasets. On the biggest (and therefore most realistic) dataset, PEANUT is 2 times faster than the best competitor (BWA-MEM). Second, PEANUT in all-mode is 4 to 10 times faster than the all-mapper RazerS 3 and 4 to 6 times faster than the all-mapper MrFast. While Bowtie 2 provides an all-mode, too, it did not terminate in competetive time due to extensive memory requirements exceeding the capabilities of the test system.

Table 1 Performance of PEANUT and other read mappers on the human reference genome on four datasets as defined in the text.

Dataset sizes are given in gigabasepairs (Gbp; 1 Gbp is 10 million reads of length 100). Run times are listed for three consecutive repetitions. Dashes indicate that no run times could be obtained due to crashes.

	Mapper	Type	Time (min:sec)	
Dataset 1 (0.5 Gbp)	PEANUT	Best-stratum	1:51	1:51	1:53	
BWA-MEM	Best	3:35	3:20	3:16	
Bowtie 2	Best	5:13	5:12	5:12	
NextGenMap	Best	3:06	3:08	3:06	
CUSHAW3	Best	9:06	9:07	9:07	
CUSHAW2-GPU	Best	2:38	2:38	2:39	
PEANUT	All	22:26	22:37	22:42	
RazerS 3	All	200:13	200:12	199:55	
MrFast	All	103:04	106:28	107:45	
Dataset 2 (0.5 Gbp)	PEANUT	Best-stratum	1:40	1:45	1:41	
BWA-MEM	Best	1:58	1:57	1:57	
Bowtie 2	Best	3:30	3:14	3:12	
NextGenMap	Best	2:28	2:38	2:29	
CUSHAW3	Best	7:44	7:45	7:44	
CUSHAW2-GPU	Best	2:20	2:22	2:21	
PEANUT	All	13:43	13:57	13:57	
RazerS 3	All	91:02	90:38	89:38	
MrFast	All	77:11	77:51	77:27	
Dataset 3 (1 Gbp)	PEANUT	Best-stratum	3:17	3:15	3:13	
BWA-MEM	Best	4:56	4:51	4:44	
Bowtie 2	Best	8:20	8:18	8:20	
NextGenMap	Best	4:46	4:42	4:45	
CUSHAW3	Best	76:38	76:25	76:29	
CUSHAW2-GPU	Best	6:30	6:09	6:07	
PEANUT	All	28:05	28:11	28:11	
RazerS 3	All	150:59	151:13	151:36	
MrFast	All	–	–	–	
Dataset 4 (10 Gbp)	PEANUT	Best-stratum	18:22	18:36	18:31	
BWA-MEM	Best	36:46	36:33	36:35	
Bowtie 2	Best	54:38	54:22	55:51	
NextGenMap	Best	–	–	–	
CUSHAW3	Best	390:20	390:15	390:41	
CUSHAW2-GPU	Best	30:23	30:30	30:34	
PEANUT	All	254:43	254:49	254:19	
RazerS 3	All	900:27	901:33	900:50	
MrFast	All	–	–	–	

Table 2 Performance of PEANUT on two alternative test systems with a four years old NVIDIA Geforce 580 GPU and a NVIDIA Tesla K40c GPU.

Dataset	Type	Time (min:sec)	
		Geforce 580	Tesla K40c	
1	Best-stratum	2:05	2:06	2:08	2:04	2:05	2:02	
All	24:04	24:33	24:50	24:01	24:24	24:26	
2	Best-stratum	1:56	1:43	1:47	1:50	1:45	1:42	
All	14:31	14:32	14:26	14:51	14:50	14:46	
3	Best-stratum	3:37	3:20	3:18	4:00	3:49	3:48	
All	29:55	30:06	30:13	30:09	30:17	30:23	
Notes.

See Table 1 for explanations.

The accuracy of the obtained alignments is assessed using Rabema (see “Validation”). For best-mappers, using the simulated dataset 1, we measure precision and recall with Rabema as defined by Siragusa, Weese & Reinert (2013): Recall is the fraction of reads correctly mapped to their original location. This location is known from the read simulation with Mason (see above). Precision is the fraction of correctly mapped reads among all reads that were mapped unambiguously (i.e., for which the mapper only provided exactly one hit). Figures 5 and 6 shows the results with increasing maximum error rate of the reads. Except when restricting to reads with zero errors (there, CUSHAW3 is slightly better), PEANUT slightly outperforms all other best-mappers in this benchmark. For best-mapping, the GCAT9 benchmark provides an alternative, less formal approach to measure the accuracy of a mapper by testing whether a mapped read is within 5 base pairs of its known origin in given datasets simulated with DWGSIM.10 For comparison, we provide GCAT results in Table 3 for the four datasets (1) 100bp-se-small-indel,11 (2) 100bp-se-large-indel,12 (3) 100bp-pe-small-indel13 and (4) 100bp-pe-large-indel,14 consisting of about 12 million 100 basepair reads each. The comparison is limited to those of our considered read mappers for which public results are available in GCAT (i.e., BWA-MEM, Bowtie 2, CUSHAW2 and CUSHAW3). The full reports are available via the GCAT web service under the URLs provided in the footnotes.

Figure 5 Recall for different best-mappers.

The obtained recall for different best-mappers given a maximum error rate.

Table 3 Alternative GCAT benchmark results for the GCAT datasets 1–4 (see text).

Dataset	Mapper	Correctly mapped	Incorrectly mapped	Unmapped	
1	PEANUT	95.07%	4.662%	0.266%	
BWA-MEM	97.47%	2.515%	0.014%	
Bowtie 2	93.52%	5.284%	1.192%	
2	PEANUT	94.78%	4.899%	0.323%	
BWA-MEM	97.34%	2.648%	0.014%	
Bowtie 2	93.37%	5.372%	1.258%	
3	PEANUT	98.34%	1.654%	0.01%	
BWA-MEM	99.22%	0.783%	0.00005%	
Bowtie 2	95.19%	3.723%	1.089%	
CUSHAW3	99.06%	0.935%	0.00002%	
CUSHAW2	99.05%	0.948%	0.006%	
4	PEANUT	98.08%	1.888%	0.035%	
BWA-MEM	99.08%	0.922%	0.00005%	
Bowtie 2	95.05%	3.806%	1.143%	
CUSHAW3	98.91%	1.089%	0.003%	
CUSHAW2	98.62%	1.091%	0.294%	

All-mappers are compared by their ability to find all alignments of a given error rate. Here, we again use the sensitivity provided by Rabema (see “Sensitivity”). This involves creating a gold standard with RazerS 3 configured to full sensitivity, which is computationally expensive. We therefore perform this on only 1,000 reads simulated with the same parameters as dataset 1. The gold standard is calculated for an error rate of at most 15%. Figure 7 shows that up to an error rate of 4% PEANUT provides a sensitivity of almost 100%, similar to RazerS 3 and nearly as good as MrFast. Beyond 5%, the sensitivity of PEANUT is supreme to RazerS 3 and MrFast. This is due to RazerS 3 and MrFast being restricted to low error rates per default to achieve acceptable performance. In summary, at default settings, PEANUT provides similar or even better sensitivity than RazerS 3 and MrFast while being 4–10 times faster (see above).

Figure 6 Precision for different best-mappers.

The obtained precision for different best-mappers given a maximum error rate.

While PEANUT requires at least 2.5 GB of GPU memory for filtration and validation, it is not restricted to running on high-end GPU models like the Geforce 780 used above: Table 2 shows that an advantage can be maintained when benchmarking on a different test system with an Intel Core i7-2600 (3.4 GHz, 16 GB RAM) and a four year old NVIDIA™ Geforce 580 GPU. We see that the older system is about 9% slower and still faster than the best competitor in Table 1 on the newer test system. In Table 2, we further provide run times for the same system equipped with an NVIDIA Tesla K40c GPU with 12 GB RAM. While the Tesla is supreme in theory, here, it cannot outperform the Geforce. This is because a faster CPU (or more CPU cores) would be needed in order to also speed up the postprocessing which runs in parallel to the GPU based filtration and validation (see “The PEANUT algorithm”).

Evaluation of mapping qualities

Finally, we evaluate the mapping qualities provided by PEANUT. For each PHRED-scaled mapping quality q, the corresponding probability 10−q10 equals the expected false positive rate. Here, true positives are the correctly identified true sampling positions of the reads, whereas false positives are reported mapping locations that may have the same alignment score but are not the true origins of a read. Figure 8 shows the measured and expected false positive rate (i.e., the fraction of false positives among all hits) at increasing mapping qualities for the hits reported by PEANUT in all-mode on dataset 1. As can be seen, the measured false positive rate rapidly decays when increasing the mapping quality from zero. A mapping quality above 10 (i.e., a probability of 0.1) already guarantees almost no false positives. Comparison with the dashed line, depicting the expected false positive rate for each mapping quality, shows that the mapping qualities provided by PEANUT are indeed conservative in the sense of underestimating the true probabilities. The parameters λ and C of the approximation (see “Postprocessing”) can be configured to different values to achieve a closer fit for a particular reference or dataset. Despite that, the defaults provide a reasonable way to distinguish between true and false positives without the computational overhead needed for calculating concrete alignments of many suboptimal hits (see “Postprocessing”).

Figure 7 Sensitivity for different all-mappers.

Sensitivity to find all alignments of a read given a maximum error rate for different all-mappers with default parameters.

Figure 8 Mapping quality versus measured false positive rate for the hits reported by PEANUT in all-mode on the simulated dataset 1.

The dashed line depicts the false positive rate as it would be expected from the probabilities encoded by the PHRED scaled mapping qualities.

Discussion

We presented the q-group index, a novel data structure for read mapping, along with parallel algorithms for index building and querying. The algorithms fit nicely to the GPU architecture by using a combination of element-wise and prefix scan operations over large arrays and requiring hardly any data transfer between the host and the GPU. We showed that the q-group index needs significantly less memory in practical scenarios than a conventional q-gram index while maintaining constant access time. The q-group index has been adopted by NVIDIA™ and is provided as part of the NVBIO library.15

On top of the q-group index we implemented the read mapper PEANUT. The q-group index enables the mapper to be the first implementation of the filtration and validation approach that works completely on the GPU. The benefit of this is illustrated by PEANUT showing supreme speed over state of the art read mappers like BWA, Bowtie 2, RazerS 3, CUSHAW3 and MrFast, as well as the GPU based read mappers NextGenMap and CUSHAW2-GPU. This holds for both the high-end GPU NVIDIA™ Geforce 780 and a four years old NVIDIA™ Geforce 580. PEANUT is 4–10 times faster than other all-mappers. Further, it is faster than all other best mappers; in particular, it is 2 times faster than the fastest best-mapper on the most realistic dataset. The speed improvements do not come at the cost of mapping quality. In fact, PEANUT is even slightly more sensitive than other all-mappers at default parameters and provides an at least comparable precision and recall compared to other best-mappers in our Rabema-based benchmarks.

PEANUT is distributed under the MIT license as an open source Python16 software package. Filtration and validation were implemented in OpenCL, using the PyOpenCL package (Klöckner et al., 2012) and postprocessing was implemented in Cython (Behnel et al., 2011). Documentation and installation instructions are available at http://peanut.readthedocs.org, where we further provide a Snakemake (Köster & Rahmann, 2012) workflow of all analyses conducted in this work.

Additional Information and Declarations

Competing Interests

Author Contributions

1 https://www.khronos.org/opencl

2 Hierarchical data format version 5, http://www.hdfgroup.org/HDF5.

3 http://www.illumina.com

4 As provided by the Rabema data package, http://www.seqan.de/projects/rabema/, October 2013.

5 http://ftp.ensembl.org/pub/release-74/fasta/homo_sapiens/dna/

6 Available at the European Nucleotide Archive under http://www.ebi.ac.uk/ena/data/view/ERR281333, 07/2014.

7 Available at the European Nucleotide Archive under http://www.ebi.ac.uk/ena/data/view/ERR091787, 07/2014.

8 http://www.illumina.com/platinumgenomes, 07/2014.

9 http://www.bioplanet.com/gcat, 08/2014.

10 https://github.com/nh13/DWGSIM, 08/2014.

11 http://www.bioplanet.com/gcat/reports/3499-acxvkaiakl/alignment/100bp-se-small-indel/peanut/compare-26-35, 08/2014.

12 http://www.bioplanet.com/gcat/reports/3522-twmulhzdnt/alignment/100bp-se-large-indel/peanut/compare-28-32, 08/2014.

13 http://www.bioplanet.com/gcat/reports/3500-rzzjxccatd/alignment/100bp-pe-small-indel/peanut/compare-23-18, 08/2014.

14 http://www.bioplanet.com/gcat/reports/3498-dqxwpnmjyn/alignment/100bp-pe-large-indel/peanut/compare-87-25, 08/2014.

15 http://nvlabs.github.io/nvbio/

16 http://www.python.org

The authors declare there are no competing interests.

Johannes Köster conceived and designed the experiments, performed the experiments, analyzed the data, contributed reagents/materials/analysis tools, wrote the paper, prepared figures and/or tables, reviewed drafts of the paper.

Sven Rahmann wrote the paper, reviewed drafts of the paper, analyzed the method mathematically.

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
