# Peer review of "Massively parallel read mapping on GPUs with the q-group index and PEANUT"

_PeerJ, doi:10.7717/peerj.606_

## Round 0.1 · original submission · Major Revisions

Authors have described a GPU-based NGS short-read aligner, which uses a data structure called as "q-group index" for the input reads. Subsequently this aligner aligns reference sequences to the reads on the GPU with the help of the q-group index. Overall, I see the strength of this manuscript, and I give a major revision decision to address critical points raised by the reviewers. Addressing these comments are essential for testing performance, accuracies and usabilities of this reader mapper. This will improve the overall strength of this manuscript.

Reviewer 1 ·

Basic reporting

No comments.

Experimental design

No comments

Validity of the findings

No comments

Additional comments

This paper presents a GPU-based next generation sequencing short-read aligner, which builds a q-group index for the input reads and then align reference sequences to the reads on the GPU with the help of the q-group index. This aligner is shown to be faster than existing CPU-based and GPU-based aligners with comparable alignment accuracy. This paper is well organized and written. The following list my comments.
Major comments:
(1) The measure the alignment accuracy, the authors have used the Rabema benchmark, which allows comparing mapping results based on a formalized framework. However, the gold-standard alignments used in Rabema (visit the homepage for more details http://www.seqan.de/projects/rabema) are constructed by aligning reads to the reference genome using the Razers 3 aligner. I do not know how accurate the Razers 3 aligner is able to align reads to a reference genome, but feel less confident on such evaluations. Since all of the reads are simulated by Mason and thus we have already had a good standard alignment file in SAM format for the evaluation, why do the authors re-align the reads using Razers 3 and then consider the reported alignments as the good standard? I would not accept such evaluations.
(2) GCAT does provide a good and fair evaluation for all aligners in terms of alignment and SNP calling. I would strongly recommend the authors to evaluate your aligner using the benchmarks of GCAT. Suggested datasets are “100bp-se-small-indel”, “100bp-pe-small-indel”, “100bp-se-large-indel”, “100bp-pe-large-indel”. Since GCAT only considers one alignment per read, the authors can only evaluate the performance of your best-mapping variant.
Minor comments:
(1) Page 1, missing the reference to “BWT”.
(2) Page 2, missing the reference to “BWA-MEM”.
(3) Page 2, “datastructure” should be changed to “data structure”.
(4) Page 3, we usually use the term “streaming multiprocessor”, rather than “symmetric multiprocessor” for CUDA-enabled GPUs.
(5) Page 3, the expression “Each SM executes one thread block until all threads in the block are completed. Once an SM has completed a thread block, it moves to the next if any blocks are left” is wrong. The correct one could be “One thread block is executed on one SM and continues resident until all threads in the block are completed. Once one thread block on an SM has been finished, another thread block will be scheduled to the SM if any blocks are left.”
(6) Page 4, “wheter” should be changed to “whether”?
(7) Page 10, for sequence alignment, percent identity is usually calculated by dividing the number of symbol matches in the optimal alignment by the optimal alignment length. Hence, I would suggest using another term, e.g. “similarity score”, instead of “percent identity”.

·

Basic reporting

The article is well written and the concepts are clearly explained.

Experimental design

The research is well executed.

The only minor problem is 2.1 where it is too general (like text book) and the authors should explain more, specifically on how they tailor the software to fit the GPU hardware architecture(s) and how it make use of the latest technology to improve the computation efficiency.

Validity of the findings

I would be interested in seeing how the software performs with the professional Tesla GPUs, which should be accessible through NVIDIA GPU test drive programme:

http://www.nvidia.com/object/gpu-test-drive.html

Given that CUDA 6 is only out in the last couple of months, many programs (stables) may not be able to catch up with the latest library. The authors should at least attempt to run Soap3dp with the older CUDA 5 and add it to the comparison.

Reviewer 3 ·

Basic reporting

The authors introduce a new data structure called the q-group index and a read mapper that uses this data structure implemented for GPUs. The q-group index is derived from the q-gram index. Instead of a direct access using the number representation of a q-gram, the q-gram partitions the possible values of the q-grams and achieves a compression through this.

The paper is generally well-written and the ideas are clearly presented. The idea of the q-group index is a good one and appears to be well-suited for using GPUs for read mapping.

Experimental design

I have two major remarks:

(1) In their experiments, the authors do not state whether and how they masked the genome. Thus, it is hard to estimate how the performance of the program is in relation to programs that do not use masking. Further, the human genome consists of 50-66% of repeats and these regions are of increasing interest and not considered as "genomic black matter" any more.

In the case that the authors use repeat masking in their benchmarks, they should repeat the benchmarks without repeat masking and present the full results, including the results without the repeat masking. For practitioners, these are the more relevant numbers.

(2) The authors should use run the Rabema benchmark with more than 1000 reads. At least 10k or 100k reads should be used to give a more reliable estimate.

I have one minor remark:

(3) The authors could partition the input to RazerS 3 and MrFast for their fourth data set to obtain numbers for these programs.

Validity of the findings

The authors should resolve (1) to show that their method also works for the relevant case of repeats. In the case of filtering repeats, their method would have an advantage over the other methods and it is not clear whether their good results are due to repeat filtering or their algorithmic advances.

Point (2) should also be resolved to increase the robustness of their results using the Rabema benchmark.

---

## Round 0.2 · accepted · Accept

We are pleased to inform you that this MS is suitable for publication, however, please go through a minor typo issue raised by reviewer #3.

Reviewer 1 ·

Basic reporting

No comments.

Experimental design

No comments

Validity of the findings

No comments

Additional comments

All of my comments have already been addressed.

·

Basic reporting

No Comments

Experimental design

No Comments

Validity of the findings

No Comments

Reviewer 3 ·

Basic reporting

All my issues have been addressed.

Note that in English, sentences continue in lower case after a colon ":" to the best of my knowledge.

Experimental design

Experimental Design looks good.

Validity of the findings

Findings are valid.